# Effectiveness of interventions designed to mitigate the negative health outcomes of informal caregiving to older adults: an umbrella review of systematic reviews and meta-analyses

Mariam Kirvalidze ![ORCID] ,[1] Ahmad Abbadi ![ORCID] ,[1] Lena Dahlberg ![ORCID] ,[1,2] Lawrence B Sacco ![ORCID] ,[3] Lucas Morin ![ORCID] ,[4,5] Amaia Calderón-Larrañaga ![ORCID] [1,6]

LM and AC-L contributed equally.

For numbered affiliations see end of article.

**Correspondence to**
Mariam Kirvalidze;
mariam.kirvalidze@ki.se

## ABSTRACT

**Objectives** This umbrella review aimed to evaluate whether certain interventions can mitigate the negative health consequences of caregiving, which interventions are more effective than others depending on the circumstances, and how these interventions are experienced by caregivers themselves.

**Design** An umbrella review of systematic reviews was conducted.

**Data sources** Quantitative (with or without meta-analyses), qualitative and mixed-methods systematic reviews were included.

**Eligibility criteria** Reviews were considered eligible if they met the following criteria: included primary studies targeting informal (ie, unpaid) caregivers of older people or persons presenting with ageing-related diseases; focused on support interventions and assessed their effectiveness (quantitative reviews) or their implementation and/or lived experience of the target population (qualitative reviews); included physical or mental health-related outcomes of informal caregivers.

**Data extraction and synthesis** A total of 47 reviews were included, covering 619 distinct primary studies. Each potentially eligible review underwent critical appraisal and citation overlap assessment. Data were extracted independently by two reviewers and cross-checked. Quantitative review results were synthesised narratively and presented in tabular format, while qualitative findings were compiled using the mega-aggregation framework synthesis method.

**Results** The evidence regarding the effectiveness of interventions on physical and mental health outcomes was inconclusive. Quantitative reviews were highly discordant, whereas qualitative reviews only reported practical, emotional and relational benefits. Multicomponent and person-centred interventions seemed to yield highest effectiveness and acceptability. Heterogeneity among caregivers, care receivers and care contexts was often overlooked. Important issues related to the low quality of evidence and futile overproduction of similar reviews were identified.

**Conclusions** Lack of robust evidence calls for better intervention research and evaluation practices. It may

## STRENGTHS AND LIMITATIONS OF THIS STUDY

⇒ The umbrella review methodology enabled us to synthesise and describe the state of the evidence on the topic of interventions to mitigate the negative health consequences of informal caregiving.
⇒ The review benefits from the mixed-methods approach, as we included both quantitative reviews on effectiveness and qualitative syntheses exploring complex aspects related to the experiences of caregivers.
⇒ Synthesis is confined to a descriptive, narrative output due to heterogeneity of included reviews.
⇒ More recent primary studies on new interventions were not captured, as they would not have been included in systematic reviews selected for this umbrella review.

be warranted to avoid one-size-fits-all approaches to intervention design. Primary care and other existing resources should be leveraged to support interventions, possibly with increasing contributions from the non-profit sector.

**PROSPERO registration number** CRD42021252841; BMJ Open: doi:10.1136/bmjopen-2021-053117.

## INTRODUCTION

Informal caregivers are defined as any relative, partner, friend or neighbour who provides a broad range of assistance to an older person who lives with a chronic or disabling condition and with whom they have a significant personal relationship.[1] Their role has become increasingly important, as populations age and professional social care services struggle to meet the increasing care demands. Improved life expectancy leads to more years spent with late-life dependency,[2] and this burden often falls on the families of older adults. It is now estimated that informal caregivers contribute to the majority of care

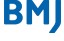

for persons aged over 50 years in most European countries,[3] and long-term care relies heavily on the availability of informal caregivers, especially in the current context of care worker shortages.[4] In the near future, some countries are expected to have 'care gaps' or, in other words, insufficient numbers of informal caregivers to meet the increasingly complex care needs of older adults.[5 6]

Informalisation of care has come with important concerns about the potential side effects of caregiving, and with growing interest from decision-makers and stakeholders in the effectiveness of existing support interventions to alleviate these negative consequences. The burden of informal care can be determined by various factors, such as the intensity of caregiving, care receivers' dependency level, relationship with the care receiver, available social support from the community, and caregivers' own health.[7–10] Research findings on the health effects of caregiving have been mixed, with some studies reporting a positive association with health and longevity,[11] while others reveal a myriad of adverse repercussions, such as increased mortality,[12] poorer well-being,[13] and worse physical and mental health outcomes in general.[14] The socioeconomic status, age and gender of caregivers have also been found to moderate these adverse consequences: women, older people, and those with lower income bear the highest psychological and physical burden of informal caregiving.[15] There is now a broad consensus about the multifactorial nature of this burden and its complex ramifications in terms of health and well-being.[15 16]

There has been a considerable increase in the number of empirical studies (both qualitative and quantitative) on the topic of caregiver support interventions, followed—in recent years—by a large number of systematic reviews. Consequently, it has become challenging to keep a bird's-eye view of this field of research. To provide decision-makers and stakeholders with synthetic and actionable evidence, a logical next step is to conduct a review of systematic reviews, that is, an 'umbrella review'.[17] Umbrella reviews are designed to give a broad and high-level overview of the available evidence on a given topic by compiling existing reviews rather than aggregating findings from the individual studies included in these reviews. By stitching together reviews about different types of interventions, populations or conditions, they provide an opportunity to assess not only the content-related comprehensiveness of these reviews, but also the overall reliability of the available evidence. Although a handful of umbrella reviews or meta-reviews have already examined caregiver support interventions, their scopes have been limited to only one type of intervention and/or disease,[18 19] and some were too broad or unsystematic to capture differences between caregiver groups.[20 21] Our understanding of the effectiveness of existing interventions for mitigating the harms of informal caregiving is currently hampered by the lack of up-to-date synthesis of the evidence focusing on more objective physical and mental health outcomes rather than perceived burden or subjective well-being. Moreover, there is a need for integrating quantitative findings about the effectiveness of interventions with qualitative findings on the lived experience of caregivers who received these interventions.

In particular, the following research questions were addressed: (1) Are there effective interventions to prevent and reduce the negative health consequences of informal caregiving?; (2) Are certain types of interventions more effective than others?; (3) Is there evidence that the effectiveness of interventions depends on caregiver, care receiver, care context and implementation characteristics?; (4) How are the proposed interventions experienced by caregivers in terms of effectiveness and implementation outcomes?

## METHODS

### Protocol registration

The protocol for this umbrella review was registered a priori in the PROSPERO database[22] and published in a peer-reviewed journal (doi:10.1136/bmjopen-2021-053117).[23] Online supplemental table 1 includes a list of all amendments made to the protocol after its registration, which will be mentioned in relevant sections below. The reporting of this umbrella review is based on the Preferred Reporting Items for Overviews of Reviews statement.[24]

### Eligibility criteria, information sources and search strategy

Inclusion and exclusion criteria used to select reviews are presented in box 1 and discussed in detail in the umbrella review protocol.[23] Due to the abundance of published systematic reviews, the eligibility criteria were amended to exclude reviews of critically low quality based on our risk-of-bias assessment. This amendment is documented in online supplemental table 1. The Medline, CINAHL, PsycINFO and Web of Science databases were initially searched from 1 January 2000 to 26 March 2021. In addition, we performed a manual search of the reference lists of included reviews. Online supplemental table 2 includes detailed search strategies and number of hits from all searched databases. Search strategies were developed by the review team in consultation with university librarians. Even though the first reviews on caregiver intervention research were published in the 1990s, we focused on reviews published since 2000 to capture studies conducted in the context of more current social settings. We also restricted the inclusion to reviews published in languages spoken by research team members: English, Swedish, Spanish, French, Italian and German. The titles and abstracts of all references as well as full texts of preliminarily selected reviews were screened against the eligibility criteria independently by two reviewers using the Covidence software developed by the Cochrane collaboration.[25] Any dissent in abstract screening and/or full-text assessment was resolved by discussion moderated by a third reviewer. Finally, our search was updated to

---

**Box 1    Inclusion and exclusion criteria for quantitative and qualitative reviews**

**Inclusion criteria**
*Publication type, date and language*
⇒ Reviews published in a peer-reviewed journal.
⇒ Reviews published between 1 January 2000 and 26 March 2021.
⇒ Reviews published in English, Swedish, Spanish, French, Italian or German.
*Study design*
⇒ For quantitative reviews: reviews including a reproducible, systematic search strategy, AND clearly defined inclusion/exclusion criteria AND risk of bias assessment for all included primary studies.
⇒ For qualitative reviews: reviews including a reproducible, systematic search strategy AND defined inclusion/exclusion criteria.
*Population*
⇒ Reviews concerning informal caregivers (ie, people who regularly provide unpaid care to a family member, friend or neighbour) of older people OR of persons presenting with ageing-related disease (eg, dementia, stroke, Parkinson's disease, cancer, heart failure, multimorbidity, frailty).
*Intervention*
⇒ Reviews focusing on interventions and assessing either their effectiveness (for quantitative reviews) or their implementation and/or the lived experience of the target population (for qualitative reviews).
*Outcome*
⇒ Reviews including physical or mental health-related outcomes of informal caregivers, including health-related quality of life.

**Exclusion criteria**
*Study quality and overlap*
⇒ For quantitative reviews: reviews of 'critically low' quality as per AMSTAR-2 assessment tool.
⇒ For qualitative reviews: reviews with two or more critical flaws as per ad hoc assessment tool.*
⇒ Review pairs with very high or high overlap (as per Corrected Covered Area method) were examined, and older, less relevant or lower-quality reviews were excluded.
*Additional exclusion criteria*
⇒ Reviews of interventions focusing *exclusively* on care receivers as the target population.
⇒ Reviews focusing *exclusively* on interventions for caregivers of young populations.
⇒ Reviews measuring *exclusively* non-health-related outcomes, such as caregiver burden, stress/strain, work or financial status, family relations, breakdown of informal care.
⇒ Reviews focusing *exclusively* on end-of-life care interventions.

*See the published protocol (doi:10.1136/bmjopen-2021-053117) for the ad hoc assessment tool for qualitative reviews.

---

capture any eligible review published between 26 March 2021 and 31 January 2023.

### Selection of reviews: risk-of-bias assessment and overlap assessment

Each quantitative review was critically appraised by two reviewers working independently, using the AMSTAR-2 checklist.[26] This checklist categorises the quality of the reviews based on seven 'critical' and nine 'non-critical' domains. The research team made a consensus-based decision to downgrade item #10 (*Did the review authors report on the sources of funding for the studies included in the review?*) from 'critical' to 'non-critical' since this information was only available in Cochrane reviews (amendment is documented in online supplemental table 1). Based on our appraisal, the reviews were grouped into 'critically low', 'low', 'moderate' and 'high-quality' categories, and critically low-quality reviews were excluded from the umbrella review. Qualitative reviews were also assessed independently by two team members, who used an ad hoc quality appraisal checklist adapted from the Joanna Briggs Institute Checklist for Systematic Reviews and Research Syntheses[17] and available in the published protocol.[23] The tool was developed and piloted by all team members on a sample of five randomly selected reviews. Items #3, #4, #7

and #10 were considered as 'critical', with reviews exhibiting more than two critical flaws being excluded. The quality of mixed-methods reviews was assessed using the above-mentioned tools for their quantitative and qualitative components, respectively. Any dissent in the risk-of-bias assessment process was resolved through discussions moderated by a third investigator.

Reporting biases arising from primary studies included in the systematic reviews were assessed using the AMSTAR-2 tool,[26] which includes critical items related, among other aspects, to selective reporting of outcomes and publication bias. We were more lenient with qualitative reviews (or parts of reviews related to qualitative data) because risk-of-bias assessment is less common and structured in qualitative research. We did, however, include items related to the quality of primary studies in our ad hoc tool for qualitative reviews.

The degree of overlap of primary studies included in the reviews (namely, the fraction of evidence synthesised in two or more reviews) was estimated using the Corrected Covered Area methodology[27] and the Graphical Representation of Overlap for OVErviews open-access tool.[28] We built separate citation matrices for quantitative and qualitative primary studies and we accounted

for structural missingness based on publication date (ie, primary papers published after the review was completed were not marked as missing from the reviews). Guided by the methodology developed by Pollock et al,[29] two team members went through the pairs with 'very high' (≥15%) and 'high' (10%–15%) overlap, and decisions on inclusion were made based on relevance, search dates, potential contribution to the umbrella review and quality of the publication (described in online supplemental table 3).

### Data extraction

Data from included reviews were extracted by two team members in structured spreadsheets designed and validated a priori by all the investigators involved in this umbrella review. The quality and validity of the extracted data were assessed through regular cross-checks. For quantitative reviews, we extracted the following information: review objectives, methodological aspects (inclusion/exclusion criteria, search dates and databases, synthesis methods), target population and disease, characteristics of the interventions of interest, sociodemographic background of caregivers, health outcomes, degree of effectiveness of interventions (with/without meta-analysis), implications for practice and research, as well as the full list of primary studies included in the review. The latter was extracted for the purpose of overlap assessment. Qualitative reviews were characterised in terms of methodology, and we extracted all the verbatim text related to the reported caregivers' experiences of interventions.

### Synthesis methods

For quantitative reviews, we conducted a narrative synthesis and provided findings in a tabular format, organised by intervention type. To provide a data-driven list of intervention types, we adapted Gaugler et al's[30] original typology into the following classification: case management, psychosocial and education/skills-building, respite care, relaxation and leisure, and mindfulness. Definitions of intervention types can be found in the latter study.[30] If reviews compared different types of interventions, the outcome of this comparison was reported. Factors related to the caregiver, care receiver and/or care context affecting the effectiveness of interventions were identified and reported whenever possible.

For qualitative reviews, we followed the 'mega-aggregation framework' synthesis method.[31] Online supplemental table 1 documents post-protocol amendments related to synthesis of qualitative reviews, while online supplemental table 4 describes steps involved in the mega-aggregation process. First, the extracted verbatim texts from reviews were coded line by line (complete coding) by two team members. Codes were then iteratively and deductively categorised based on the adapted version of van Houtven et al's framework.[32] Throughout the review, we refer to verbatim texts extracted from reviews and their corresponding codes as third-order constructs, while primary studies inform second-order constructs, and the communication from participants (ie, caregivers)

is considered as a first-order construct. Thus, the themes, categories and subcategories that we generated based on third-order constructs (ie, reviews) are termed as fourth-order constructs.[31] In case our fourth-order codes did not fit fully into the framework, they were inductively categorised into new themes. Codes referring to care receiver or staff opinions were removed. A third investigator checked the outcome of the categorisation independently, and several consensus meetings were held to resolve disagreements and finalise the synthesis of qualitative materials.

As a complementary output, we used parallel convergence approach (ie, synthesising quantitative and qualitative evidence separately and bringing them together at the final stage) to update our initial conceptual framework that was based exclusively on expert opinion and/or existing literature (see the published protocol[23]). The resulting framework, From Support Interventions to Improved Caregiver Outcomes (SIICO), will be presented below and aims to substantiate the potential pathway (and its various modifiers and mediators) linking caregiver support interventions to improved health outcomes.

### Patient and public involvement

No patients or members of the public were involved in the development of this umbrella review. However, the scope and methods of this review were informed by the literature and discussions with experts in the field.

## RESULTS

### Selection of reviews: risk-of-bias assessment and overlap assessment

Our search strategy resulted in a total of 6209 unique records, of which 5906 were excluded at the stage of title and abstract screening (figure 1). From the 303 reviews that underwent full-text screening, 158 were excluded and 145 were further assessed for risk of bias. We excluded 92 quantitative and 9 qualitative reviews considered as being of 'critically low' quality. Results of the risk-of-bias assessment for all potentially eligible reviews are reported in online supplemental table 5A,B.

Of the 51 remaining quantitative reviews, 32 fell under the 'low quality' category, 13 were classified as being of 'moderate quality' and 6 were rated as high-quality reviews. Overall, the most common issues were lack of pre-registered protocol, absence of a full list of excluded studies with a rationale for the exclusion of each study, lack of reporting of sources of funding for primary studies, methodological issues related to meta-analyses, and suboptimal assessment of heterogeneity and publication bias. All 18 qualitative reviews eligible at this stage had only one or no critical flaw. Most common pitfalls included: lack of clarity regarding participation of researchers in synthesis process, lack of justification of qualitative synthesis methods and lack of information on data extraction procedures.

After risk-of-bias assessment, reviews were examined for primary study overlap. Full citation matrices (accounting

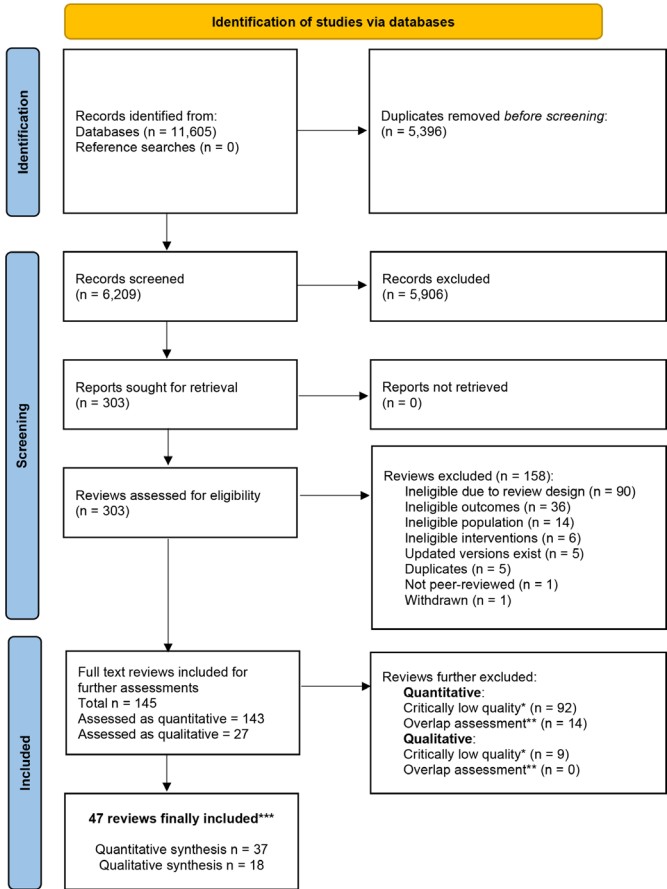

**Figure 1** Preferred Reporting Items for Systematic Reviews and Meta-Analyses 2020 flow diagram.[86] *Quantitative reviews with AMSTAR-2 category of 'critically low' quality were excluded; qualitative reviews with two or more critical flaws were excluded. **Overlap assessment was completed for quantitative and qualitative reviews separately and in distinct subgroups based on support intervention types. ***Out of 47 distinct reviews, 10 contributed to qualitative data only, 8 were included as both qualitative and quantitative data sources, while 29 reviews contributed exclusively to quantitative synthesis.

for structural missingness based on publication date) and resulting pyramids are available in online supplemental files 1 and 2. Among the 18 qualitative and mixed-methods reviews assessed for overlap, we did not exclude any reviews since the degree of overlap was low. However, out of the 51 eligible quantitative reviews, we examined 33 pairs that reported 'very high' level of overlap (≥15%) and further checked 40 pairs of reviews that reported 'high' level of overlap (10%–15%). Upon rigorous assessment of overlap, 14 quantitative reviews were finally excluded (see online supplemental table 3 for detailed description of the overlap decisions).

Finally, 47 reviews covering 619 primary studies were included in the umbrella review. Of these, 10 contributed to qualitative data only, 8 were included as both qualitative and quantitative data sources, while 29 reviews contributed exclusively to quantitative synthesis.

## Characteristics of the included systematic reviews

All included reviews were published in English. Publication dates spanned from 2009 to 2021. Most primary studies were conducted in North America or Western Europe. Reviews set various inclusion and exclusion criteria regarding the design of primary studies, with some reviews focusing on randomised controlled trials, while others casted a wider net including quasi-experimental and qualitative studies. Among quantitative reviews, 18 undertook a meta-analysis while 19 undertook a data synthesis without meta-analysis. In qualitative reviews (or qualitative parts of mixed reviews), the most common synthesis approach was narrative (14 reviews), but four reviews reported a qualitative meta-synthesis. Characteristics of all included reviews, such as search dates, number of studies, demographics of included populations, and types of interventions examined are summarised and presented in online supplemental table 6.

## Main findings: quantitative reviews

### Case management interventions

Eight reviews reported quantitative findings related to the effectiveness of the interventions involving case management (table 1 and online supplemental table 7).[33–40] Seven of these reviews focused on caregivers of persons with dementia,[33–39] while one took interest in caregivers of stroke survivors.[40] Health-related outcomes varied and included: depression (n=6), health-related quality of life (n=6), anxiety (n=2), general health (n=1) and self-rated health (n=1). Three reviews included a meta-analysis.[35 38 40] Five reviews reported no significant effect of case management interventions on caregivers' health outcomes,[34 35 37 39 40] one review provided inconclusive findings,[33] one review reported one primary study showing positive effects on depression,[36] and finally another review[38] showed a significant improvement in short-term depressive symptoms and general health (although with waning effects at longer-term follow-ups). Two reviews demonstrated that multicomponent interventions that include other types of approaches in addition to case management may have positive effects.[34 35] Lee et al[35] also reported that interventions including caregivers with high quality of life at baseline or those who cared for people with less serious health conditions were less likely to be effective. The other five reviews either did not explore the issue of heterogeneity of caregivers, care receivers and care context, or stated that these data were not reported by primary studies.[40]

### Psychosocial and education/skills-building interventions

A total of 23 reviews reported quantitative findings related to effectiveness of interventions involving psychosocial support, education and skills-building for caregivers (tables 2 and 3 and online supplemental table 8). Of these, 19 reviews included and synthesised findings on both psychosocial and educational interventions,[34 35 37 39–54] while two reviews focused solely on psychosocial support,[55 56] and another two only on

**Table 1** Interventions involving case management

| First author, year | Included primary studies | Disease of care receivers | Characteristics of intervention | Control group | Health outcomes of caregiver | Quality of evidence |
|---|---|---|---|---|---|---|
| Goeman, 2016[33] | 24 RCTs, 1 cohort study, 1 case study, 2 mixed methods | Dementia | Case management by nurse or dementia care provider involving visits, calls and emails | Usual care, home visits, educational sessions, phone calls, helpline or not reported | Depression, HR-QoL, general health | ●●●● |
| Greenwood, 2016[39] | 1 quasi-experimental | Dementia | General practice-based intervention delivered by a nurse, to augment care from primary care physicians | Usual care and educational materials | Depressive symptoms | ●●○○ |
| Hopwood, 2018[34] | 2 RCTs, 1 quasi-experimental, 7 mixed methods | Dementia | Internet-based interventions involving nurses, health professionals, social workers using various internet-based media to manage caregivers | Face-to-face delivered information, usual care | Depression, anxiety, HR-QoL, self-rated health | ●●○○ |
| Lee, 2020[35] | 4 RCTs | Dementia or MCI | Community-based interventions delivered by various professionals. Duration varied from 3 to 18 months | Usual care or not reported | HR-QoL | ●●○○ |
| Lucero, 2019[36] | 2 RCTs | Dementia | Telephone or computer-based health-related planning, monitored by nurses | Usual care or training on available local resources | Depression, anxiety | ●●●○ |
| Piersol, 2017[37] | 2 RCTs | Dementia | Case management by nurses or occupational therapists | Not reported | HR-QoL | ●●○○ |
| Reilly, 2015[38] | 11 RCTs | Dementia | Dyadic and non-dyadic interventions involving a case manager. Majority face-to-face, delivered by various professionals. Duration varied from 4 months to 2 years | Waiting list, usual care or augmented usual care | HR-QoL, depression, general health | ●●●● |
| Pucciarelli, 2020[40] | 5 RCTs, 1 quasi-RCT | Stroke | Informational interventions that included home visits and management after discharge | Usual care or not reported | Depression, HR-QoL | ●●●○ |

Legend for quality assessment based on AMSTAR-2: ●●●● high; ●●●○ moderate; ●●○○ low.
A summary of the main findings of each review is available in online supplemental table 7.
HR-QoL, health-related quality of life; MCI, mild cognitive impairment; RCT, randomised controlled trial.

education/training.[57 58] Care receivers' health problems spanned from dementia (n=12) and stroke (n=4) to cancer (n=4) and other chronic illnesses (n=3). Most reviews included multiple interventions (eg, information provision, psychosocial support, educational training, skills-building), even though the label 'multicomponent' was used differently across reviews. Meta-analysis was performed in 14 reviews. Of these, seven reviews reported insignificant effect estimates or little to no effects following psychosocial and/or educational interventions,[35 40 41 44 48 54 59] four reviews reported significant effect estimates related to similar interventions,[47 52 53 58] while

**Table 2** Psychosocial, psychoeducational and skills-building interventions: reviews that included both psychosocial and educational interventions (n=19)

| First author, year | Included primary studies | Disease of care receivers | Characteristics of intervention | Control group | Health outcomes of caregiver | Quality of evidence |
|---|---|---|---|---|---|---|
| Akarsu, 2019[47] | 13 RCTs | Dementia | Psychological, multicomponent and educational interventions. Delivered to ethnic minority caregivers | Minimal support measures | Depression | ●●●○ |
| Gonzalez-Fraile, 2021[48] | 26 RCTs and quasi-RCTs | Dementia | Remotely delivered interventions only. Predominant components: training with or without information, support with or without information, and interventions including both support and training elements | Usual treatment or waiting list, minimal support, providing information only | Depression, depressive symptoms, HR-QoL | ●●●○ |
| Greenwood, 2016[39] | 2 RCTs, 1 quasi-RCT | Dementia | Education, training, CBT, delivered for various duration and locations (home-based, primary care) | Usual care or not reported | Depression, general health | ●●○○ |
| Hopwood, 2018[34] | 9 RCTs, 20 mixed-methods, 7 quasi-RCTs | Dementia | Internet-based only. Information, online sessions or modules, links to resources, training, peer interaction online, small groups peer support. Duration varied from 2 weeks to 12 months | Information only, waiting list, email newsletter, usual care, telephone support, written information, video, website | HR-QoL, depression, anxiety, self-rated health | ●●○○ |
| Lee, 2020[35] | 14 RCTs | Dementia or MCI | CBT, group sessions, dyadic sessions, home visits, meetings, website, support calls | Usual care or information only | HR-QoL | ●●○○ |
| Lins, 2014[52] | 9 RCTs | Dementia | Telephone counselling with or without educational material and workbook. Varying methods and duration (20–60 min per call) | Usual care or friendly calls | Depression, anxiety | ●●●○ |
| Piersol, 2017[37] | 41 RCTs and quasi-RCTs | Dementia | Group interventions, CBT, single-component support interventions, multicomponent psychoeducational interventions. Delivered through various methods | Not reported | HR-QoL, depression, anxiety | ●●○○ |
| Teahan, 2020[53] | 24 RCTs | Dementia | Counselling, information, education on dementia, CBT, relaxation techniques, communication skills, emotional control, other skills, MBSR, physical exercise, dyadic or individual | Usual care, educational material, follow-up calls, enhanced respite care | Depression, HR-QoL, general health | ●●○○ |
| Wiegelmann, 2021[50] | 37 RCTs and quasi-RCTs | Dementia | Psychoeducation, counselling, CBT and peer support. Delivered either face-to-face or remotely | Usual care or not reported | HR-QoL, depression | ●●○○ |
| Zabihi, 2020[49] | 14 RCTs | Dementia, other illnesses | Behavioural activation, education, group support interventions, among others | Usual care, waiting lists, phone calls, educational interventions | Depression (symptoms and diagnosis) | ●●○○ |

Continued

**Table 2** Continued

| First author, year | Included primary studies | Disease of care receivers | Characteristics of intervention | Control group | Health outcomes of caregiver | Quality of evidence |
|---|---|---|---|---|---|---|
| Gabriel, 2020[42] | 6 RCTs, 6 NRSIs | Cancer | CBT, coping theory, psychoeducation, skills-building | Usual care | Psychological/emotional and physical domains of QoL | ●●○○ |
| Heckel, 2019[43] | 2 RCTs | Cancer | Telephone helplines. Variable duration, number of outcalls and content of the calls | Usual care | Depression, emotional distress | ●●○○ |
| Treanor, 2019[44] | 21 RCTs and quasi-RCTs | Cancer | Psychoeducational approaches in the form of coping skills training. Delivered predominantly by nurses. Most face-to-face, some by telephone or video | Usual care or information only | HR-QoL, depression, anxiety, emotional distress, physical health status | ●●●○ |
| Waldron, 2013[45] | 6 RCTs | Cancer | Skills training and CBT. Dyadic or individual. Some interventions delivered face-to-face, some by telephone | Usual care | HR-QoL | ●●○○ |
| Corry, 2019[41] | 21 RCTs | Various | Psychosocial, educational and psychoeducational interventions. All interventions were individual and telephone based | Usual care or non-telephone-based support | HR-QoL, psychological health (depression, anxiety), physical health | ●●●○ |
| Sin, 2018[46] | 26 RCTs and quasi-RCTs | Various | Web-based ICT interventions (at least part of an intervention had to be web based). Varying content and duration (from several days to months) | Usual care or not reported | Depression, anxiety, HR-QoL | ●●●○ |
| Forster, 2012[51] | 7 RCTs | Stroke | Informational interventions. Delivered in varying formats and duration | No information, usual care, educational material, waiting list, workbook | Depression, HR-QoL | ●●●● |
| Minshall, 2019[54] | 8 RCTs | Stroke | Individual, group or dyadic interventions. Delivered by various professionals, in person or through telephone. Duration varied from 1 month to 3 years | Usual care or not reported | Depression, HR-QoL | ●●●○ |
| Pucciarelli, 2020[40] | 6 RCTs and quasi-RCTs | Stroke | Educational interventions, with components of psychoeducation. Some delivered face-to-face, some by phone/web, some through home visits | Usual care or not reported | Depression | ●●●○ |

Legend for quality assessment based on AMSTAR-2: ●●●● high; ●●●○ moderate; ●●○○ low.
A summary of the main findings of each review is available in online supplemental table 8.
CBT, cognitive–behavioural therapy; HR-QoL, health-related quality of life; ICT, information and communication technology; MBSR, mindfulness-based stress reduction; MCI, mild cognitive impairment; NRSI, non-randomised study of intervention; QoL, quality of life; RCTs, randomised controlled trials.

three reviews reported significant changes in mental health specifically after cognitive–behavioural therapy interventions.[49 55 56] Narrative reviews often failed to reach definitive conclusions due to discordant results (similar numbers of studies reporting significant and non-significant estimates),[42 43 45 46 50 51] but some reviews reported overall positive findings.[34 37] Conclusions were discordant regarding remote interventions. For instance, the evidence documenting the benefit of telephone-based psychosocial support was found to be inconclusive

**Table 3** Psychosocial, psychoeducational and skills-building interventions: reviews that included only psychosocial (n=2) or only educational (n=2) interventions

| First author, year | Included primary studies | Disease of care receivers | Characteristics of intervention | Control group | Health outcomes of caregiver | Quality of evidence |
|---|---|---|---|---|---|---|
| Bennett, 2019 [58] (occupational therapy) | 9 RCTs | Dementia | Occupational therapy delivered at home for dyads. Tailored and goal-oriented interventions | Usual care, education, collaborative call | Emotional distress, HR-QoL | ●●○○ |
| Smith, 2019 [57] (training) | 19 RCTs and quasi-RCTs | Stroke and older adults | Training delivered to facilitate care after discharge. Varying delivery modes (face-to-face, by phone, by different professionals) and duration | Usual care, information only | HR-QoL, depression, anxiety | ●●●○ |
| Hopkinson, 2019 [56] (psychosocial) | 25 RCTs and quasi-RCTs | Dementia | CBT. Varying delivery modes and duration | Support group control, psychoeducation control, information support control, usual care | Depression, anxiety | ●●○○ |
| Wang, 2020 [55] (psychosocial) | 6 RCTs | Neurocognitive diseases | Bibliotherapy. Either web or video based. Varying number of sessions and duration of each session | Usual care, waiting list, educational video | Depression, anxiety | ●●●○ |

Legend for quality assessment based on AMSTAR-2: ●●●● high; ●●●○ moderate; ●●○○ low.
A summary of the main findings of each review is available in online supplemental table 8.
CBT, cognitive–behavioural therapy; HR-QoL, health-related quality of life; RCTs, randomised controlled trials.

by Corry et al[41] and Gonzalez-Fraile et al,[48] while Lins et al[52] reported positive effects on depressive symptoms. Lee et al[35] and Hopwood et al[34] grouped certain interventions under the 'multicomponent' category and reported that these were associated with significant positive effects on health-related quality of life and anxiety and depression, respectively. Treanor et al[44] reported significant positive effects on quality of life in the short term, with waning effects over time. Akarsu et al[47] examined the effectiveness of psychological, multicomponent and educational interventions in ethnic minority caregivers, reporting an overall mean reduction in depression scores.

Some reviews explored findings from primary studies depending on the caregiver, care receiver and care context characteristics as well as implementation-related aspects. Successful interventions were reported to be more individualised,[45] proactive rather than reactive,[51] developed using user input from the target groups,[39 47] and guided by competent professionals[37] or peer caregivers with relatable experiences.[35] Heckel et al[43] reported that telephone helplines were mostly used by white, higher-income, middle-aged women, and lack of participation from other groups of caregivers should be investigated. Corry et al[41] acknowledged that these data are often not reported by primary studies.

## Respite care

Four reviews explored the effectiveness of respite care services (table 4 and online supplemental table 9).[60–63] Three of these reviews focused on caregivers of persons with dementia,[61–63] while one review took interest in caregivers of people with any chronic illness.[60] Health-related outcomes varied and included: depression (n=4), anxiety (n=2) and general health (n=1). The largest review in this category—and the only one including a meta-analysis— reported no significant effects on caregivers' mental health outcomes.[60] However, narrative findings from studies not eligible for meta-analysis in the same review and another review by Maffioletti et al[61] were rather positive, although they remained discordant.[60 61] Vandepitte et al[62] and Maayan et al[63] reported small or insignificant effects in relation to caregivers' health outcomes. Maayan et al[63] reported that care receivers' disease severity could have been positively correlated with the effectiveness of respite, with caregivers of patients with milder symptoms not requiring as many breaks. The price of respite care (if privately purchased) was identified as an important factor for effectiveness and access.[60] Shaw et al[60] found that longer interventions tended to have stronger benefits than shorter ones, and that the short-term incidence

**Table 4** Interventions involving respite care

| First author, year | Included primary studies | Disease of care receivers | Characteristics of intervention | Control group | Health outcomes of caregiver | Quality of evidence |
|---|---|---|---|---|---|---|
| Maayan, 2019[63] | 3 RCTs | Dementia | In-home respite, day care, institutional care. Delivered by trained carers as well as volunteers | Usual care, waiting list | Depression | ●●●● |
| Maffioletti, 2019[61] | 10 quasi-experimental, 4 cross-sectional | Dementia | Self-financed or paid day care services by professionals or volunteers. Caregivers were supported in some studies with music therapy, socialising or just free time | Usual care or not reported | Depression, psychological well-being, general health | ●●○○ |
| Vandepitte, 2016[62] | 5 RCTs and NRSIs | Dementia | Day care, in-home delivery of respite. Varying duration and frequency | No respite | Depression, anxiety | ●●○○ |
| Shaw, 2009[60] | 26 RCTs and quasi-RCTs, 79 observational studies | Various | Day care, mixed interventions, in-home and institutional care | Usual care or no respite | Depression, anxiety | ●●○○ |

Legend for quality assessment based on AMSTAR-2: ●●●● high; ●●●○ moderate; ●●○○ low.
A summary of the main findings of each review is available in online supplemental table 9.
NRSI, non-randomised study of intervention; RCTs, randomised controlled trials.

of depression was reduced among people who received home respite care but not in trials that evaluated day care.

## Relaxation and leisure

A total of six reviews were grouped under this category (table 5 and online supplemental table 10),[35 37 64–67] including ones focusing on caregivers to people with dementia (n=3), neurological diseases (n=1) and various chronic illnesses (n=2). Health-related outcomes varied and included: depression (n=4), health-related quality of life (n=4), anxiety (n=4), physical health (n=3), and blood pressure and weight (n=1). Two reviews included a meta-analysis.[67 68] Interventions were heterogeneous: two reviews included interventions related to physical activity/exercise,[64 65] one review included relaxation/complementary medicine interventions,[35] one review focused on creative arts interventions,[66] while two other reviews focused on 'miscellaneous' interventions, including exercise, leisure programmes and relaxation.[37 67] These interventions had generally positive effects. Meta-analyses by both Lee et al[35] and Cheng et al[67] showed large positive effect sizes for the effect of relaxation and miscellaneous activities on health-related quality of life and depressive symptoms. Reviews with narrative synthesis reported that creative arts interventions tended to have positive effects on well-being,[66] while exercise programmes resulted in lower blood pressure and less depressive symptoms,[64] as

well as increased muscle mass, strength[65] and better physical health.[37]

Most of the reviews in this category did not consider heterogeneity associated with caregiving factors or implementation characteristics. According to Doyle et al,[65] spousal and family caregivers may gain more from engaging in dyadic exercise compared with when their care receivers exercise independently. Miscellaneous interventions examined by Cheng et al[67] showed that caregivers' mean age significantly moderated the intervention effects, whereby younger caregivers benefited more in terms of reducing depressive symptoms.

## Mindfulness-based interventions

Four reviews included interventions using mindfulness-based stress reduction techniques (table 6 and online supplemental table 11).[67 69–71] Three reviews focused on caregivers of persons with dementia,[67 69 70] while one review took interest in caregivers of people with various illnesses.[71] Health-related outcomes varied and included: depression (n=3), anxiety (n=3), and cognition and biomarkers for stress (n=1). Two reviews included meta-analysis.[67 69] Mindfulness-based interventions showed significant positive effects for reducing depressive symptoms and anxiety levels immediately after interventions, but the effects were largely attenuated at follow-up.[69 70] Cheng et al[67] reported a significant positive

**Table 5** Interventions involving relaxation, physical activity or leisure

| First author, year | Included primary studies | Disease of care receivers | Characteristics of intervention | Control group | Health outcomes of caregiver | Quality of evidence |
|---|---|---|---|---|---|---|
| Cheng, 2020[67] | 12 RCTs | Dementia | Miscellaneous: physical activity, spiritual care, complementary therapies (eg, religious activities, expressive writing) | Usual care or alternative sessions | Depression, anxiety, HR-QoL, physical health | ●●●○ |
| Lee, 2020[35] | 2 RCTs | Dementia or MCI | Complementary medicine including yoga, massage and meditation | No treatment or respite care | HR-QoL | ●●○○ |
| Piersol, 2017[37] | 3 experimental studies | Dementia | Exercise programme, adapted leisure programme, night-time monitoring system | Not reported | Physical health | ●●○○ |
| Irons, 2020[66] | 8 pre/post-trials | Neurological diseases | Creative arts interventions, such as music, drama, dance, song writing | Usual care or not reported | HR-QoL, anxiety, depression | ●●○○ |
| Cuthbert, 2017[64] | 9 RCTs | Various | Physical activity interventions: walking, aerobics, yoga. Varying intensities and formats | Waiting list, usual care or not reported | Depression, anxiety, well-being, physical strengthening, blood pressure, weight | ●●○○ |
| Doyle, 2020[65] | 5 RCTs, 6 quasi-RCTs | Various | Physical activity dyadic (DyEx) and non-dyadic (DySplit) interventions | DyEx vs DySplit (ie, exercising together or not) | Depression, anxiety, physical health | ●●○○ |

Legend for quality assessment based on AMSTAR-2: ●●●● high; ●●●○ moderate; ●●○○ low.
A summary of the main findings of each review is available in online supplemental table 10.
HR-QoL, health-related quality of life; MCI, mild cognitive impairment; RCTs, randomised controlled trials.

effect on depression in grouped meta-analysis. Parkinson et al[71] reported mixed results, with interventions showing some positive changes in anxiety and depression scores, but with small and waning effects. While some authors acknowledged the importance of potential moderating factors and heterogeneity among caregivers,[70 71] there were no findings to report regarding this question.

### Comparison across categories of interventions

Three reviews compared the effect sizes across different types of interventions.[47 67 68] Lee et al[68] provided a separate meta-analysis on multicomponent interventions (including social support, education and skills-building), which showed the largest positive effect on health-related quality of life compared with single-component interventions. Similarly, Cheng et al[67] reported that multicomponent and miscellaneous interventions had the strongest

effects on depressive symptoms. However, Akarsu et al[47] reported that effect sizes across multicomponent, psychological and educational interventions are broadly similar.

### Main findings: qualitative reviews

For the qualitative part of our umbrella review, we extracted and analysed data from 18 reviews. Online supplemental table 4 describes the steps undertaken to apply mega-aggregation framework synthesis[31] to our data. The results of our convergent synthesis of quantitative and qualitative evidence are presented in figure 2. Two themes were identified based on the findings from the 18 reviews providing qualitative data: (1) intervention outcomes and (2) implementation outcomes. These two themes reflect two main domains of intervention implementation research.[72] A detailed list of all fourth-order constructs, accompanying the third-order constructs

**Table 6** Interventions involving mindfulness-based activities

| First author, year | Included primary studies | Disease of care receivers | Characteristics of intervention | Control group | Health outcomes of caregiver | Quality of evidence |
|---|---|---|---|---|---|---|
| Cheng, 2020[67] | 7 RCTs | Dementia | MBSR and its modifications | Usual care or alternative sessions | Subjective well-being, depression | ●●●○ |
| Liu, 2018[69] | 5 RCTs | Dementia | MBSR and its modifications | Usual care or active comparison (respite, social support) | Depressive symptoms, anxiety | ●●●● |
| Shim, 2020[70] | 9 RCTs | Dementia and MCI | MBSR and its modifications | Usual care or active comparison (psychoeducation, music listening) | Cognition, depression, mindfulness, anxiety, biomarkers for stress | ●●○○ |
| Parkinson, 2019[71] | 1 RCT, 5 quasi-RCTs | Various | MBSR and its modifications | Usual care or not reported | Anxiety | ●●○○ |

Legend for quality assessment based on AMSTAR-2: ●●●● high; ●●●○ moderate; ●●○○ low.
A summary of the main findings of each review is available in online supplemental table 11.
MBSR, mindfulness-based stress reduction; MCI, mild cognitive impairment; RCTs, randomised controlled trials.

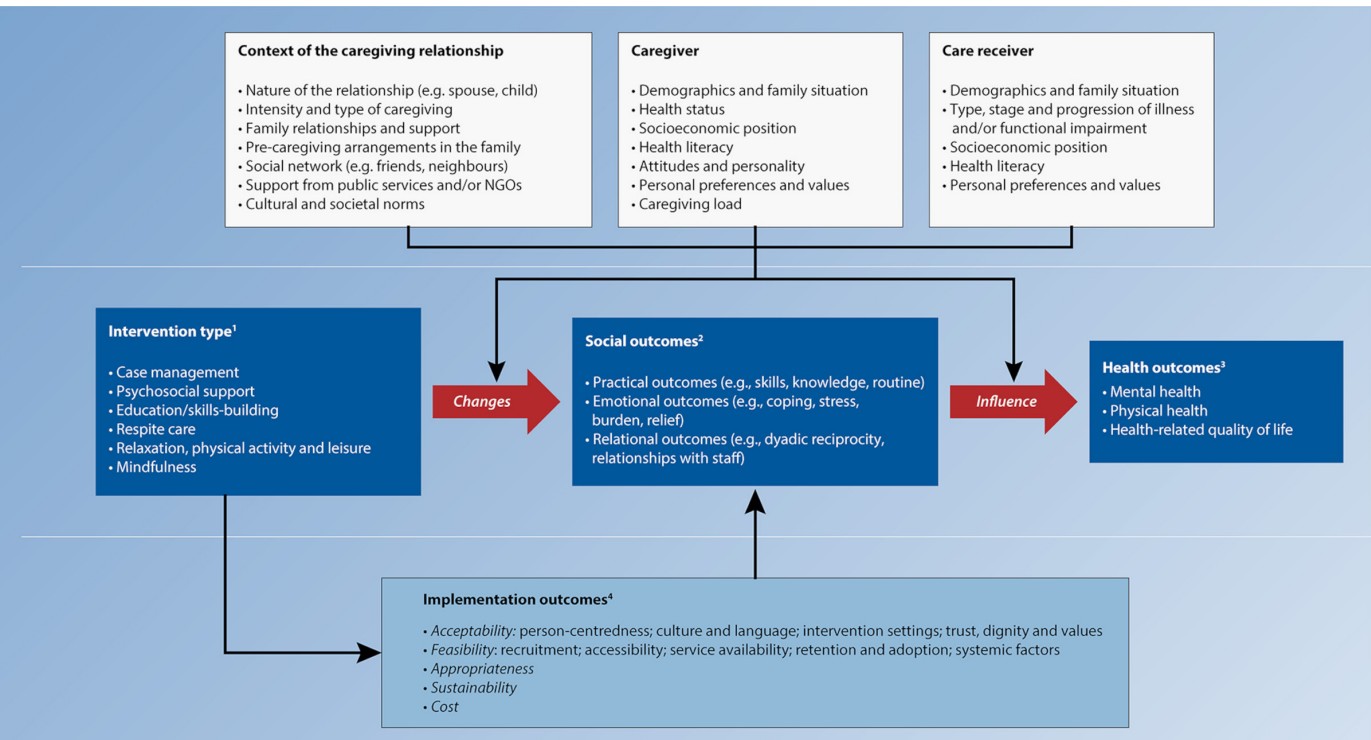

**Figure 2** From Support Interventions to Improved Caregiver Outcomes framework. The list of potential modifying factors and sources of heterogeneity of intervention effects, such as relationship context, caregiver and care receiver characteristics, were inspired by van Houtven et al's[32] organising framework for caregiver interventions and amended by review authors based on expert opinion and review of broader caregiving literature (boxes *Context of the caregiving relationship, Caregiver* and *Care receiver*). [1]The typology of interventions by Gaugler et al[30] was reorganised in a data-driven fashion and currently includes six types of interventions (box *Intervention type*). [2]Based on qualitative findings of our umbrella review, there is evidence that interventions affect practical, emotional and relational aspects of caregivers' lives (box *Intervention social outcomes*). [3]According to mainly quantitative data, some of the interventions might also have effects on caregivers' mental health, physical health and health-related quality of life (box *Intervention health outcomes*). [4]Intervention effects are also influenced by implementation outcomes, which were selected based on Proctor et al's[79] outcomes for implementation research, and derived from qualitative data, as expressed by caregivers (box *Implementation outcomes*). NGOs, non-governmental organisations.

(verbatim text) and references is available in online supplemental table 12.

## Theme 1: intervention outcomes

This theme described the potential benefits and/or harmful effects to caregivers from participating in the interventions (ie, efficacy and effectiveness), and was informed by 13 reviews.[33 46 52 59 60 66 71 73–78] Only two reviews[71 73] reported health-related benefits, namely decreased depression and anxiety[71 73] and better quality of life.[73] Twelve reviews documented on social outcomes of interventions, that is, outcomes related to the caregiver's social life, day-to-day routine and relationship with care receivers and other people.[33 46 52 59 60 66 73–78] Based on these reviews, we divided social outcomes into three subcategories: practical, emotional and relational outcomes.

Practical outcomes were explored in nine reviews[33 52 59 60 66 73–76] examining how support interventions could introduce changes in the caregivers' day-to-day lives. According to caregivers, interventions were effective in providing them with new knowledge that enhanced coping, ensured better management of burden and improved their caregiving skills.[59 73 74] Acquired skills increased their readiness for care, allowed them to build strategies to solve problems and gave them a sense of normality.[52 66 73 74 76] Additionally, respite care was found by Shaw et al[60] to 'give structure to the carer's week along with a sense of normality' as the caregiver was able to 'match the ebb and flow of caregiving activities'. However, occasionally, there were additional burdens related to respite care use: 'many hassles […] involved in the preparation for respite care' that would lead to loss of 'the physical and emotional energy'.

Participation in the interventions also brought about emotional outcomes, which were described by nine reviews.[46 52 60 66 73 75–78] Across several reviews, caregivers reflected on the positive impacts of talking about their daily arduous challenges, sharing experiences or simply providing a moment to escape from their various duties through singing, arts or just chit-chat.[52 66 76 78] These positive impacts included gaining a sense of relief, better coping with stress, enjoyment, reduced social isolation and gaining emotional support.[46 52 66 76–78] Lins et al[52] highlighted that even the caregivers in the control group of the intervention, where participants had 'conversations only about general topics such as the weather, television, movies, news or social activities', reported the social interactions to be a 'helpful alternative to relieve carers'. However, some reviews described negative emotional consequences to the participation in interventions. Shaw et al[60] argued that the physical break due to respite was 'not sufficient in itself to provide the mental break that was needed by most carers to improve their well-being'. It would require that the participants had 'total disengagement from the caring role'.[60] Moreover, they described that the use of respite care can be perceived as selfish by the caregiver and bring about feelings of guilt.[60] Irons et

al[66] asserted that, while creative arts interventions may increase positive feelings, negative feelings might not be completely removed.

Finally, four reviews investigated the relational outcomes of support interventions, namely how these interventions modified the caregivers' relationships with their care receiver or peers.[66 75–77] Dyadic interventions were reported to promote increased engagement and deepened relationships between the caregiver and the care receiver.[66 76] The practice of creative arts like singing, viewing art, writing music or creating memory albums brought a sense of reciprocity, fostered better communication, allowed for seeing the care receiver under a new light and improved quality time.[66 76] Du Preez et al[75] also reported increased engagement between caregivers and care receivers after returning from respite day care services.

## Theme 2: implementation outcomes

A total of 17 included reviews examined to what extent the adoption of effective interventions was determined by implementation-related barriers and facilitators that could render them ineffective in certain circumstances.[72 79] Following the classification of Proctor et al[79] and Hull,[80] we synthesised the findings from these reviews in terms of (1) acceptability, (2) feasibility, (3) appropriateness, (4) sustainability and (5) implementation costs.

### *Acceptability*

Although the included reviews covered studies with varying designs and heterogeneous interventions, we found similarities in the description of the determinants of perceived caregiver acceptability. First, seven reviews spanning almost all intervention types demonstrated that the *person-centeredness, flexibility* and *personalisation* of the interventions promoted higher acceptability as expressed by caregivers.[33 34 46 60 66 81 82] Caregivers appreciated if interventions were able to accommodate their lifestyles and needs.[33 34 46 66 81] Some caregivers might prefer face-to-face meetings compared with online meetings,[33] while others might appreciate a self-paced programme.[46] Additionally, caregivers expressed more favourability towards interventions when the deliverer considered not only their preferences, but also those of the care receiver, including their medical conditions, availability and commitment level.[33 46 60 66 82] In a review focused on respite,[60] caregivers highlighted accepting or rejecting the interventions through the lens of care receivers, and whether aspects ranging from their physical and mental health to their cooperation and approval of participation were fully considered by deliverers. Manifestations of lack of patient-centredness affected care receivers not only during the period of respite, but also after the respite, as it took some time for care receivers to recover. Along the same lines, Miles et al[82] indicated that caregivers were positive towards the use of patient and caregiver information and support services but suggested that 'there is not a one-size-fits-all approach which can be used, as every

patient [...] and their carers will have different needs, preferences and responses'.

Second, three reviews documented the importance of cultural and linguistic aspects in explaining the success of support interventions,[33 52 60] in particular whether interventions were able to accommodate the needs and values of different ethnic and religious caregiver groups. Language was a barrier for those who did not speak the national language that the intervention was delivered in.[52] This prevented minorities and those who did not have a full grasp of the language from participating, especially if the intervention required advanced understanding and usage of the language, such as counselling.[52] Lins et al also explained that 'receiving counselling in the native language was also shown to contribute to building trust'. Additionally, having multicultural staff facilitated the acceptability of the interventions because better relationships could be built with the families, and the needs of the different communities could be better understood.[33 60] For example, in the use of in-home respite, minority groups would prefer working with same gender and ethnicity personnel, who also spoke the same language.[60] Ensuring the appropriateness of food based on religious restrictions, and being mindful of cultural and religious differences were also highlighted in the same review.[60]

Third, six reviews examined how the physical, social and structural characteristics of interventions promoted or hindered their acceptability by informal caregivers and care receivers.[46 60 66 74–76] Settings that facilitated communication between caregivers and receivers seemed to provide caregivers with a feeling of having a 'special place' and thus encouraged their engagement with the intervention.[66 74] On the other hand, online-based interventions were sometimes perceived by caregivers as implying technical difficulties and hindering rapport with the staff.[46] In other words, caregivers expressed their preference for certain delivery methods, as they better suited their needs. For example, in-home respite care was seen as less disruptive compared with day care.[60] Moreover, high rates of staff turnover were reported to disrupt the continuity of care.[60] Du Preez et al[75] asserted caregiver concerns related to a lack of knowledge about activities taking place at respite day care: 'family carers have little to no contact with the adult day service other than to ready their care recipient for the day's attendance and have little knowledge of how their care recipient spends their time while attending adult day service'.

Finally, five reviews built on qualitative findings to address the role of trust, dignity and ethical values in the degree of acceptability of support interventions.[52 60 66 75 81] Caregivers seemed to appreciate the intervention of staff who they knew well, and were reluctant to accept the advice from those who they had never met.[52 81] Additionally, caregivers appraised the emotional attitude of the staff and the latter's investment in their case.[52] The more familiar the staff was with the case and needs of the caregivers, the more appreciated the intervention

was.[52 81] Moreover, respecting the privacy of caregivers and receivers[60 75] and treating care receivers with care and dignity were especially valued by caregivers.[52 60 66 75] The moral values embodied by the staff were also important for caregivers.[66]

### Feasibility

Intervention feasibility is defined by Proctor et al[79] as 'the extent to which a new intervention can be successfully used or carried out within a given setting. [...] It is invoked as a potential explanation of an intervention's success or failure, as reflected in recruitment, retention, or participation rates.' Feasibility of support interventions was explored in 13 of the 18 qualitative reviews included in our umbrella review. We further divided this category into five dimensions: (1) recruitment, (2) accessibility, (3) availability, (4) adoption and retention, and (5) systemic factors.

First, challenges to *recruitment* were mentioned in four reviews.[46 60 75 81] Awareness about the availability of interventions was indeed deemed essential for ensuring participation.[60 75] Du Preez et al[75] explained that 'medical practitioners were identified as having limited knowledge of community support services and access to information resulting in poor referral processes and therefore, poor utilization by family carers and people living with dementia'. In other words, the lack of knowledge about these services at the primary care level was highlighted by caregivers across reviews.[60 75] Yet, the most preferred and accessible location for dissemination and advice on support services was precisely the primary care centre, as expressed by caregivers.[60]

Second, the feasibility of interventions was questioned in terms of their *accessibility* by six reviews,[33 46 52 60 75 76] which examined how both physical and non-physical external factors were potentially affecting the degree of accessibility of various interventions. For instance, some interventions excluded care receivers if they did not have a confirmed diagnosis,[33] while others, like singing and creative arts, included care receivers with diverse stages of disease.[76] Additionally, intervention sites were sometimes described as being 'too far away', with no available or reasonable transportation to reach the site.[60 75] In such cases, the use of alternative methods like telephone-based interventions were reported to be useful, as they avoided the hustle of transportation.[52]

Third, *service availability*—that is, the coverage of support interventions in terms of time schedules and availability of staff—was broached in six reviews.[52 60 61 76 78 81] Caregivers across several studies expressed the need for the interventions to be available outside working hours, to include weekends, and even up to 24/7 availability in the case of counselling.[52 60 76 81] The availability outside of working hours enables usage of the interventions by working caregivers.[60] Alternative measures, such as the use of answering machines outside counselling hours, were perceived as insufficient.[52]

Fourth, we found five reviews that looked at aspects related to *adoption and retention*, that is, what factors affect the initiation and the continued use of the services provided as part of an intervention.[34 46 59 60 75] The use of internet-based interventions was sometimes accompanied by technical difficulties that increased the risk of dropping out, especially among older caregivers.[34 46] Sin *et al*[46] reported that for internet-based psychoeducation interventions, 'usability problems (such as oral communication/chat quality, audio-visual function failure) were also identified as attributing to high drop-out rates (up to 50%) in some studies'. Additionally, with the progression of care receivers' disease, some caregivers reported not being able to leave the care receivers alone in respite care, leading to withdrawal from the interventions.[60 75] On the other hand, flexible, multicomponent and holistic approaches addressing the complex needs of caregivers and care receivers resulted in higher utilisation rates.[60 75]

Finally, four reviews investigated *systemic factors* (ie, health and social care system features that can affect the delivery and utilisation of support interventions).[33 73 75 81] Caregivers reported that their experience of not feeling prioritised by the staff over system-related factors hindered the use of interventions.[75] Additionally, caregivers found it difficult to retain all the information and coordinate with different practitioners and institutions.[81] They expressed the need for a central source of information to consult with.[73] Having several providers to coordinate with was linked to other problems according to caregivers, like competition for delivery or lack of involvement of the other providers.[33]

### Appropriateness
This category, which was documented in nine reviews,[33 34 52 60 66 71 73 76 82] is defined by Proctor *et al*[79] as the perceived suitability and usefulness of interventions to address the needs of caregivers. The appropriate delivery of interventions was largely dependent on the adoption of a patient-centred approach, and the existence of a multi-agency and interorganisational cooperation to address the specific needs of the caregivers.[33 34] However, it was emphasised that not all caregivers would benefit from all interventions.[34 66 71 76] As caregivers' needs and preferences differ, intervention components should be tailored to each case.[33 34 73] This referred to the type of intervention (eg, psychoeducational, relaxation, etc), the delivery mode (eg, via phone, in person), the setting (eg, at home, in the clinic) and type of participation (eg, individual, in groups, dyadic).[33 34 73] Pritchard *et al*[73] asserted that providing 'appropriate modality and timing of information' to caregivers requires 'information to be presented in different ways (eg, in writing, diagrams) repeated on several occasions and in person, not over the phone'. Additionally, some interventions such as counselling were deemed more needed during specific times, for example, during crises or in acute conditions.[52] Moreover, caregivers did not always find dyadic interventions that were effective for care receivers suited to their needs.[66 76]

### Sustainability
This category grouped those factors associated with a sustained, long-term use of the intervention[79] and was based on three reviews.[59 60 78] The sustainability of the interventions was claimed to depend on the needs and experiences of caregivers.[60] Smith and Greenwood[59] described that caregivers who had encountered peers with similar experiences were more likely to continue the peer support after the intervention had ended. The fact that mindfulness-based exercises can be practised anytime and are not limited to a certain setting seemed to facilitate its continuous use.[78] Shaw *et al*[60] reported that the opportunistic use of respite care could lead to a more regular use once its potential benefits had been experienced.

### Implementation costs
This category was linked to the financial costs associated with implementing or using the intervention[79] and was developed based on four reviews.[33 46 60 75] In general, interventions that were not provided free of charge made them less accessible, as affordability differed between individuals. This barrier was reported from reviews that included case management,[33] psychoeducational[46] and respite care interventions.[60 75]

### Search update in January 2023
The literature search was updated by the university librarians to capture reviews published between 26 March 2021 (end of our initial search) and 31 January 2023. This search yielded 1920 additional entries. A single reviewer completed title/abstract screening, leading to 57 potentially relevant reviews. After a thorough assessment of the full-text articles, a total of 26 reviews were found to meet all eligibility criteria. Finally, we excluded 14 reviews of critically low quality. The 12 remaining reviews are listed and summarised in table 7. Their key findings and recommendations are in line with the main findings of our umbrella review.

## DISCUSSION
### Summary of findings
Addressing the negative health outcomes of informal caregiving is a major challenge. In this umbrella review on support interventions for people providing informal care to older adults, we synthesised data from 47 systematic reviews covering 619 distinct primary studies. This is, to date, the most comprehensive map of the available evidence. Four main conclusions stem from our analysis.

First, whether existing interventions are effective at reducing the negative impact of caregiving on the physical and mental health of caregivers remains uncertain. Quantitative reviews provided largely discordant findings, with reviews rated as being at low risk of bias reporting trivial or no benefits.[33 38 51 63 69] Also, systematic reviews that included a meta-analysis were more likely to report a lack of effectiveness. While some case management,

**Table 7** Summary of the new reviews published between March 2021 and January 2023

| First author, year | Title | Design(s) | Included primary studies | Main findings and health outcomes |
|---|---|---|---|---|
| Andrades-Gonzalez, 2021[87] | e-Health as a tool to improve the quality of life of informal caregivers dealing with stroke patients: Systematic review with meta-analysis | Quantitative | 12 RCTs | Findings across studies are heterogeneous. However, approximately two-thirds of the studies that were part of the meta-analysis showed a decrease in depressive symptoms and a substantial improvement in the quality of life with the use of e-Health. Measures on physical health were either inconclusive or non-significant. |
| Boyt, 2022[88] | Internet-facilitated interventions for informal caregivers of patients with neurodegenerative disorders: Systematic review and meta-analysis | Quantitative | 20 RCTs, 31 pre/post evaluative studies | Internet-delivered interventions were superior in reducing anxiety, compared with controls. Findings were inconclusive for quality of life outcomes. Ten studies reported depression outcomes. The random-effects meta-analysis demonstrated that there was no significant difference between groups at post-intervention measurement. |
| Crocker, 2022[89] | Information provision for stroke survivors and their carers: Cochrane review | Quantitative | 12 new RCTs (update) | Authors are uncertain whether active information provision reduces or increases cases of carer anxiety; however, it might slightly reduce anxiety symptoms. Findings on depression are similarly inconclusive. Active information provision may have little to no effect on carer quality of life. |
| Garnett, 2022[90] | mHealth interventions to support caregivers of older adults: Equity-focused systematic review | Both | 14 experimental, 7 qualitative, 7 mixed-methods | mHealth interventions were positively received by study participants. Impacts on caregivers' mental and psychological health status were generally positive. Some participants reported challenges associated with participation; for example, interventions were too complex or difficult to understand, interventions included questions that were overly obtrusive or confronting, while some questions triggered painful memories. Some participants preferred in-person interventions. |
| Ghosh, 2022[91] | Systematic review of dyadic psychoeducational programs for persons with dementia and their family caregivers | Qualitative | 1 qualitative study involving multiple case studies | Dyadic psychoeducational programmes that were goal oriented and tailored to address individual needs had consistent benefits on various aspects of health and quality of life for the dyads. Findings on caregivers' physical and mental health outcomes were inconclusive, with similar numbers of studies reporting positive and non-significant effects. |
| He, 2022[92] | The effectiveness of multi-component interventions on the positive and negative aspects of well-being among informal caregivers of people with dementia: A systematic review and meta-analysis | Quantitative | 31 RCTs | Meta-analyses showed small to moderate effects on depression, and a moderate to high effect on caregiver anxiety. This review suggests that individualised multicomponent interventions for caregivers may be one of the ways to promote their well-being. |
| Kusi, 2022[93] | The effectiveness of psychoeducational interventions on caregiver-oriented outcomes in caregivers of adult cancer patients: A systematic review and meta-analysis | Quantitative | 28 controlled trials | Psychoeducational interventions had beneficial effects on depression, anxiety and quality of life at the immediate post-intervention period. At longer-term follow-up, the effectiveness of interventions was maintained on quality of life and anxiety, but not on depression. |
| Mårtensson, 2023[94] | Psychological interventions for symptoms of depression among informal caregivers of older adult populations: A systematic review and meta-analysis of randomized controlled trials | Quantitative | 15 controlled trials | A small effect size favouring the intervention was found for symptoms of depression, and interventions were effective in reducing incidence of major depression and psychological distress. Authors warn that, given the high heterogeneity and high risk of bias, findings should be interpreted with caution. |

Continued

**Table 7** Continued

| First author, year | Title | Design(s) | Included primary studies | Main findings and health outcomes |
|---|---|---|---|---|
| Sun, 2022[95] | Comparative efficacy of 11 non-pharmacological interventions on depression, anxiety, quality of life, and caregiver burden for informal caregivers of people with dementia: A systematic review and network meta-analysis | Quantitative | 85 RCTs | Acceptance and commitment therapy, behavioural activation, mindfulness-based intervention, multicomponent intervention, psychoeducation and cognitive–behavioural therapy might reduce depression. Notably, psychoeducation was the only effective intervention against anxiety. Only support groups had a statistically significant effect on the quality of life. |
| Thompson, 2021[96] | How singing can help people with dementia and their family care-partners: A mixed studies systematic review with narrative synthesis, thematic synthesis, and meta-integration | Both | 26 experimental, 9 qualitative, 5 mixed-methods | Results from the syntheses suggest that singing can positively impact the lives of people with dementia and their care partners, although due to heterogeneity of study design and outcome measures, it is difficult to draw conclusions based on quantitative data alone. Qualitative data provide further context and insights from participants' perspectives. For instance, participants report enjoyment, improvement in mood, social belonging and dyadic relationship. |
| Wallace, 2021[97] | Do caregivers who connect online have better outcomes? A systematic review of online peer-support interventions for caregivers of people with stroke, dementia, traumatic brain injury, Parkinson's disease and multiple sclerosis | Qualitative | 7 mixed-methods, 4 case series | Overall, participants responded positively to the psychosocial elements of the interventions. Some participants felt less lonely and more supported, while others noted that they found reading other users' posts distressing or felt that sharing their story with others was a betrayal to their family members. Participants identified convenience as a major benefit of the online platform, noting that it reduced the need to travel, take time off work or leave vulnerable family members on their own. Anonymity was identified as both a benefit and disadvantage to the use of online platforms. |
| Watt, 2022[98] | Systematic review of group-based creative arts interventions in support of informal caregivers of adults: a narrative synthesis | Qualitative | 12 qualitative, 7 mixed-methods | Positive themes emerging from qualitative data included: creative arts as unique, enjoyable and supporting expression, meaningful connection and support between caregivers, and a positive impact on dyad relationship. Some participants identified barriers related to interventions, such as emotional exhaustion, getting upset, not enough time to complete the activity, burden of caring and difficulty getting to the art gallery. |

RCTs, randomised controlled trials.

psychosocial and mindfulness interventions with more than one follow-up time point seemed to demonstrate short-term benefits, their positive effect waned as time elapsed.[38 44 69–71] Qualitative reviews provided only limited insight: although informal caregivers mentioned social and practical benefits,[52 66 73–77] they rarely spoke about how support interventions impacted their own health.[71 73]

Second, we found that multicomponent interventions showed more consistent positive effects on health outcomes across reviews, despite a large heterogeneity in what these interventions actually entailed.[34 35 67] Moreover, in the two reviews that performed meta-analyses by type of intervention, multicomponent interventions showed the largest effect sizes.[35 67] In this respect, our findings

corroborate the conclusions of two previously published umbrella reviews.[18 20]

The third conclusion is that the available evidence on support interventions relies on the simplistic assumption that informal caregivers represent a homogeneous target population, with little attention being paid to the variability in caregiver, care receiver, care context and implementation characteristics. Hence, while a number of reviews reported valuable information about the characteristics of the interventions being evaluated,[35 37 39 45 47 51] the socioeconomic and ethnic background of caregivers and care receivers and the nature of their relationship were largely overlooked. Others have emphasised the need to better account for the social determinants of health among informal caregivers.[83] Their diversity goes

beyond the obvious differences in the underlying health conditions of the persons they provide care to, and this reality should come under greater scrutiny in future studies designed to assess the effectiveness of support interventions.

Finally, our overview casts new light onto how support interventions are experienced by informal caregivers. By synthesising the qualitative findings of 18 distinct reviews, we showed that caregivers mention a myriad of social benefits, for instance, improved relationships with care receivers, better organised routine and less stress/burden associated with caregiving.[33 45 52 59 60 66 73–78] Across multiple reviews, caregivers were found to favour flexible, person-centred and needs-based interventions rather than 'off-the-rack' support services.[33 34 45 60 66 81 82] This serves to further emphasise that one-size-fits-all approaches are unwarranted since different caregivers have different preferences in terms of, among others, mode of delivery and duration of what constitute adequate support. Interestingly, we were able to find evidence across most implementation outcomes highlighted in the framework by Proctor et al[79] (acceptability, feasibility, appropriateness, sustainability and cost). However, *fidelity* of the interventions, that is, the degree to which an intervention was implemented as it was intended, was not reported.

The extensive set of qualitative findings incorporated in this umbrella review shows that support interventions targeting informal caregivers seem to improve a wide range of practical, emotional and relational outcomes. It is likely that these benefits translate indirectly into positive changes in caregivers' mental and physical health, even though high-quality evidence for this connection was lacking from quantitative reviews. To better visualise this pathway, we organised the findings from our umbrella review into a framework, SIICO, as presented in figure 2. While relations in this framework are only hypothesised and remain untested, it represents an attempt to visualise the numerous mechanisms implicated in previous research that link interventions to caregivers' health. However, our hypothetical connections between the different boxes of the framework should be interpreted with caution and deserve further scrutiny, especially concerning the potential mediating effect and transition from social and practical benefits to the improvement of objective health outcomes. It is possible that the majority of these interventions are only effective on outcomes related to health but not considered herein (eg, burden, life satisfaction, well-being) and on other aspects of caregivers' lives (eg, ability to reconcile caregiving and employment, volunteering, socialising and/or leisure). It is also possible that the observed social and relational benefits simply do not translate into measurable improvements in health outcomes, or that these improvements remain partly invisible due to methodological issues (eg, lack of statistical power, suboptimal control groups, inconsistent outcome measurements or insufficiently long follow-ups).

Beyond its initial goals, our umbrella review identified several important knowledge gaps in caregiver intervention research that, we believe, could serve as a roadmap for future studies in this field. Hence, there was a clear over-representation of certain types of interventions (ie, psychosocial interventions such as cognitive therapies, group or individual support and psychoeducation), care receiver diseases (ie, dementia) and outcomes (ie, mental health). Our umbrella review also highlights the overproduction of systematic reviews and meta-analyses on the effectiveness of support interventions for informal caregivers: more than 145 reviews have been published in the last two decades and, as expected, the overlap between them is substantial. This overproduction has become even more evident upon updating our search: between March 2021 and January 2023, a total of 57 additional reviews (compared with 303 for 2000–2021) were published. This raises questions about the potential waste of valuable research resources, especially since the overall quality of published reviews was poor. Approximately two-thirds of the reviews that we identified were rated as being of 'critically low' quality. However, one should keep in mind that some of these reviews were carried out before current guidelines and risk-of-bias appraisal tools became mainstream; in fact, more recent reviews demonstrated higher adherence to such guidelines. Qualitative synthesis and reporting practices were often subpar, with important contextual and methodological items missing from an unexpectedly large share of included reviews. The lack of consensus regarding the classification of support interventions was yet another struggle: much like Gaugler et al,[30] we found that not only were certain types of interventions described differently across primary studies, but even the same primary studies (ie, same interventions) were sometimes put into different categories from one review to another. This hinders the comparability of findings across primary studies, reviews and umbrella reviews, and makes any attempt at drawing robust conclusions about the effectiveness of interventions challenging. Others have already mentioned these incoherencies, together with other methodological shortcomings such as incomplete reporting in the included primary studies, which has an inevitable ripple effect on the degree of completeness of reviews.[20 21 84] Finally, the fact that our findings are mostly inconclusive despite an abundance of published literature highlights the methodological flaws that afflict a large number of primary studies. We believe that this should prompt a discussion between academics, stakeholders and public funding agencies. Maybe it is now time to take the advice from English methodologist Doug Altman seriously: we need less research, better research and research done for the right reasons.[85]

## Implications for public health and practice

Against the backdrop of staff shortages and budget restrictions worldwide (exacerbated by the ongoing COVID-19 pandemic), certain public health services will need to be prioritised over others. Four main implications for public health and practice emerged from our review. First, better intervention research and evaluation practices are

warranted to create an evidence base for resource allocation: we would not know what works best in different contexts if we keep relying on low-quality studies. Second, time has come for a more targeted approach to intervention design. While caregiver interventions may not work for the 'average caregiver', targeting high-risk groups—for example, caregivers with pre-existing conditions, multimorbidity or frailty—might deliver more convincing and cost-effective results. Third, since our healthcare systems are already under considerable pressure, support interventions should build on existing resources rather than entirely new services. To our surprise, very few of the interventions included in the systematic reviews and meta-analyses that we synthesised mustered primary care professionals to support informal caregivers. Yet, family physicians and home care nurses (among others) have frequent contacts with caregivers, with whom they often share a long-lasting relationship. They should have the means to remain vigilant, identify high-risk subgroups of caregivers and enlist them in relevant support programmes. Fourth, our umbrella review also highlights the potential for non-profit organisations: trained volunteers could, for instance, take a larger role in delivering social care services to alleviate the burden of caregivers.

### Implications for research

Our findings may be useful for decision-makers trying to untangle the state of the evidence on this complex topic, but also for those interested in more specific intervention types, given the broadness of our review. There are moreover several implications for future research stemming from our umbrella review. First, the enforcement of a priori protocol registration could avoid the wasteful production of reviews. Second, the consistent application of systematic review guidelines (eg, Preferred Reporting Items for Systematic Reviews and Meta-Analyses guidelines, Cochrane methods, guidance by Joanna Briggs Institute) is warranted to ensure higher quality of evidence synthesis. Third, at the primary study level, a better definition of intervention components and a clear harmonisation of intervention types are needed to ensure comparability of the generated evidence. Last, future interventions should identify and report results across subgroups defined by caregiver, care receiver and care context characteristics, and consider including not only social and relational outcomes, but also objective mental and physical health outcomes, measured comprehensively and over an extended period of time. This will be essential to better understand the potential pathways connecting social to physical/mental health outcomes, although the latter will require a mixed-methods evaluation approach.

### Strengths and limitations

The primary strength of this umbrella review is the rigorous study selection process, initiating from a generous search strategy identifying over 6000 abstracts. Such process was completed in duplicate and included quality and overlap assessment. Further, we included both quantitative and qualitative evidence to better understand the complex phenomenon of caregiver support interventions and their impact on health. In addition, data synthesis and framework elaboration were guided by existing theory and expertise within the group, adding to the rigour of the review. However, umbrella reviews are limited in terms of drawing conclusive statements, given that they do not assess primary studies, nor retest meta-analyses provided by single systematic reviews. Thus, the interpretability of our findings is limited by the quality and conclusions of included reviews, which are considerably heterogeneous across reviews. In addition, recent primary studies on new interventions may not be captured, as they may not yet have been included in systematic reviews. Finally, our umbrella review focused on caregiver-centred interventions and, thus, could not shed light on the potentially positive spillover effect of formal care services delivered to patients on informal caregivers' health. Yet, the importance of well-organised and sufficiently staffed professional services for household chores and personal care should not be underestimated, and the expansion of support services for informal caregivers cannot come to the detriment of formal care.

## CONCLUSIONS

Despite an abundance of systematic reviews, whether certain support interventions are effective at improving informal caregivers' physical and/or mental health is uncertain due to a lack of high-quality evidence. It seems that multicomponent and flexible interventions are more likely to address the complex needs of caregivers, making them more acceptable and thus leading to more tangible effects on objective health outcomes. To confirm this, we do not need more reviews: we need more carefully designed intervention studies that look at both subjective and objective health outcomes, and account for heterogeneity in caregiving.

**Author affiliations**
[1]Aging Research Center, Department of Neurobiology, Care Sciences and Society, Karolinska Institutet and Stockholm University, Solna, Sweden
[2]School of Health and Welfare, Dalarna University, Falun, Sweden
[3]Stress Research Institute, Department of Psychology, Stockholm University, Stockholm, Sweden
[4]Department of Medical Epidemiology and Biostatistics, Karolinska Institutet, Solna, Sweden
[5]Inserm CIC 1431, University Hospital of Besançon, Besançon, France
[6]Stockholm Gerontology Research Center, Stockholm, Sweden

**Acknowledgements** We wish to thank Emma-Lotta Säätelä and Jonas Pettersson, librarians at Karolinska Institutet, for preparing the search strategy and performing the searches across databases.

**Contributors** AC-L obtained the funding and drafted the protocol including the initial methodology for the umbrella review. She also acts as a guarantor of the study. AC-L, LM, MK, LD and LBS participated in further conceptualising the protocol and critically reviewing it before publication. For the umbrella review, all authors participated in study selection and quality appraisal of the reviews in duplicate. MK and AA extracted and cross-checked data from selected reviews. MK and AA drafted the initial manuscript, which was critically revised

several times by AC-L, LM, LD and LBS. The review team held several meetings regarding methodology, interpretation of data and desired output of the project. All authors gave final approval to this version of the protocol. All authors agree to be accountable for all aspects of the work in ensuring that questions related to the accuracy or integrity of any part of the work are appropriately investigated and resolved.

**Funding** This work was supported by the Swedish Research Council for Health, Working Life and Welfare (FORTE; grant number 2020-01544).

**Competing interests** None declared.

**Patient and public involvement** Patients and/or the public were not involved in the design, or conduct, or reporting, or dissemination plans of this research.

**Patient consent for publication** Not required.

**Ethics approval** This review is a tertiary synthesis of published systematic reviews (secondary synthesis). Thus, no ethical approval was sought and no contact was made with participants of primary studies included in systematic reviews.

**Provenance and peer review** Not commissioned; externally peer reviewed.

**Data availability statement** Data sharing not applicable as no datasets generated and/or analysed for this study.

**ORCID iDs**
Mariam Kirvalidze http://orcid.org/0000-0001-6773-3792
Ahmad Abbadi http://orcid.org/0000-0001-9373-668X
Lena Dahlberg http://orcid.org/0000-0002-7685-3216
Lawrence B Sacco http://orcid.org/0000-0002-4275-5378
Lucas Morin http://orcid.org/0000-0002-8486-8610
Amaia Calderón-Larrañaga http://orcid.org/0000-0001-9064-9222

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
