## [Reviewer comments · BMJ Open]

ARTICLE DETAILS

TITLE (PROVISIONAL)	Effectiveness of interventions designed to mitigate the negative health outcomes of informal caregiving to older adults: an umbrella review of systematic reviews and meta-analyses
AUTHORS	Kirvalidze, Mariam; Abbadi, Ahmad; Dahlberg, Lena; Sacco, Lawrence; Morin, Lucas; Calderón-Larrañaga, Amaia

VERSION 1 – REVIEW

REVIEWER	González-de Paz, Luis Institut d'Investigacions Biomèdiques August Pi i Sunyer
REVIEW RETURNED	16-Nov-2022

GENERAL COMMENTS	Given that interest of Informal caregivers this study gathers information from reviews. Therefore the study is of interest to researchers and clinicians. I suggest some minor revisions from the abstract. Abstract: Line 5: State the problem in a affirmative phrase (e.g. “the problem”, the state: has fragmented our understanding of the effectiveness of interventions to support informal caregivers. Line 16: state the methods and variables needed. Line 24: This is methods, but not results: “Each potentially eligible review underwent rigorous critical appraisal and citation overlap assessment”. Line 24-34: state the results of the study clearly, avoiding opinions (e.g.: Our synthesis showed that the evidence regarding the effectiveness of interventions on physical and mental health outcomes was inconclusive”. Results are evidences from the umbrella that leads you to affirm it. We found important issues related to the quality of evidence and overproduction of similar reviews. Which issues? Line 32: Some of the statemanet in the conclusions sectionas are not linked are not linked with the results. Strengths and limitations: “perform high-level synthesis”... This high level evidences should be provided in the abstract. Avoid superlatives without meaning. Narrative output: please inform about methods used in the abstract.
---

REVIEWER	Hughes, M. Courtney Northern Illinois University, School of Health Studies
REVIEW RETURNED	04-Jan-2023

GENERAL COMMENTS	This umbrella review examined reviews since 2000 that focused on caregiver health. The authors registered the review with Prospero and followed PRISMA guidelines. The authors did a great job categorizing the reviews to date and highlighting trends and important findings. The summary of findings, in particular, is well-written and include logical and insightful conclusions stemming from the review. The first line of the Objectives in the Abstract reads more like an opinion than a fact. The authors should re-word this sentence to remove the part that the reviews have “fragmented our understanding” and, instead, keep it more factual about the reviews. One point I recommend explaining further in the manuscript: I find it hard to accept the reason for not including reviews before 2000. The reason provided is “to capture studies conducted in the context of more current social settings.” It does not seem that social settings changed that much between the 1990s and 2000s. Overall, this is a helpful contribution to the literature that also guides caregiver researchers on what types of studies might be most useful going forward.
---

VERSION 1 – AUTHOR RESPONSE

Reviewer 1		
Abstract: Line 5: State the problem in an affirmative phrase (e.g. “the problem”, the state: has fragmented our understanding of the effectiveness of interventions to support informal caregivers.	We thank the reviewer for his/her comments about the abstract. We have now revised the abstract taking also into consideration the editor’s request to reformat it. We hope you will find the current version satisfactory.	Abstract
Line 16: state the methods and variables needed.		
Line 24: This is methods, but not results: “Each potentially eligible review underwent rigorous critical appraisal and citation overlap assessment”.		
Line 24-34: state the results of the study clearly, avoiding opinions (e.g.: Our synthesis showed that the evidence regarding the effectiveness of interventions on physical and mental health outcomes was inconclusive”.	We understand the reviewer’s concern regarding the statement in the results section of the abstract. We have now revised the abstract and hope our wording has improved. It is important to note that the inconclusive nature of the evidence that stems from the available literature is not the	Abstract

Results are evidences from the umbrella that leads you to affirm it. We found important issues related to the quality of evidence and overproduction of similar reviews. Which issues?	opinion of the authors, but the main result of our umbrella review. As we demonstrate in our manuscript, this is mostly due to the wide heterogeneity in the methodological strength of the studies published until know. We develop these points in the discussion, but due to limited space we cannot elaborate further in the abstract.	
Line 32: Some of the statements in the conclusions sections are not linked with the results.	We have now revised the abstract taking into consideration these comments and the editor's request to reformat it. We have tried to link the conclusion with the results and we hope you will find the current version satisfactory.	Abstract
Strengths and limitations: "perform high-level synthesis"... This high level evidences should be provided in the abstract. Avoid superlatives without meaning.	We agree with the reviewer and have reworded the statement. "The umbrella review methodology enabled us to synthesize and describe the state of the evidence on the topic of interventions to mitigate the negative health consequences of informal caregiving."	Strengths and limitations of this study
Narrative output: please inform about methods used in the abstract.	We have now clarified the limitation. "Synthesis was confined to a descriptive, narrative output due to heterogeneity of our sources." We have also included information on the methods in the abstract: "Quantitative review results were synthesized narratively and presented in tabular format, while qualitative findings were compiled using the mega-aggregation framework synthesis method."	Strengths and limitations of this study
Reviewer: 2		
The authors did a great job categorizing the reviews to date and highlighting trends and important findings. The summary of findings, in particular, is well-written and include logical and insightful conclusions stemming from the review.	We would like to thank the reviewer for a positive comment on our work. We tackled a large body of evidence indeed and hope to impact the trends in research related to informal caregiving, particularly regarding the synthesis of intervention effects.	NA
The first line of the Objectives in the Abstract reads more like an opinion than a fact. The authors should re-word this sentence to remove the part that the reviews have "fragmented our understanding" and, instead, keep it more factual about the reviews.	We have now reworded this sentence and reformatted the abstract based on the editor's request and both reviewers' feedback. "Objectives. This umbrella review aimed to evaluate whether certain interventions can mitigate the negative health consequences of caregiving, which interventions are more effective than others depending on the	Abstract

	circumstances, and how these interventions are experienced by caregivers themselves.”	
One point I recommend explaining further in the manuscript: I find it hard to accept the reason for not including reviews before 2000. The reason provided is “to capture studies conducted in the context of more current social settings.” It does not seem that social settings changed that much between the 1990s and 2000s.	This is an important point. We respectfully disagree with the reviewer’s objection to our search date limits. Since we are not limiting our search to a specific country or region, it is not easy (or possible, for that matter) to know which social/structural settings were relevant to each intervention covered by the included systematic reviews. In several countries and regions, there were important changes in social care systems and economic development towards the end of the 20th century (including, but not limited to, the shift to ambulatory care). In addition, there was a large growth of literature on ageing-in-place policies in the 2000s,¹ making the topic of informal care more relevant. Of course, it is also naïve to assume that social settings have been stable throughout 2000s, up until today, but we believe it is a safer choice. ¹Sarinnapha Vasunilashorn, Bernard A. Steinman, Phoebe S. Liebig, Jon Pynoos, "Aging in Place: Evolution of a Research Topic Whose Time Has Come", Journal of Aging Research, vol. 2012, Article ID 120952, 6 pages, 2012. https://doi.org/10.1155/2012/120952	NA
Overall, this is a helpful contribution to the literature that also guides caregiver researchers on what types of studies might be most useful going forward.	We are grateful to the reviewer for his/her time and assessment.	NA